# Use of a Fibula Free Flap for Mandibular Reconstruction in Severe Craniofacial Microsomia in Children with Obstructive Sleep Apnea

**DOI:** 10.3390/jcm12031124

**Published:** 2023-01-31

**Authors:** Krzysztof Dowgierd, Rafał Pokrowiecki, Andrzej Myśliwiec, Łukasz Krakowczyk

**Affiliations:** 1Department of Clinical Pediatrics, Head and Neck Surgery Clinic for Children and Young Adults, University of Warmia and Mazury, Żołnierska 18a Street, 10-561 Olsztyn, Poland; 2Craniofacial Center, Regional Specialized Children’s Hospital in Olsztyn, 10-561 Olsztyn, Poland; 3Institute of Physiotherapy and Health Science, The Jerzy Kukuczka Academy of Physical Education in Katowice, Ul. Mikołowska 72A, 40-065 Katowice, Poland; 4Oncological and Reconstructive Surgery Clinic, Branch of National Oncological Institute in Gliwice, Maria Sklodowska-Curie Institute—Oncology Centre (MSCI), Ul. Wybrzeze Armii Krajowej 15, 44-100 Gliwice, Poland

**Keywords:** craniofacial microsomia, obstructive sleep apnea, free fibular flaps, microsurgical reconstruction

## Abstract

This is a retrospective study describing a multi-stage protocol for the management of severe mandibular hypoplasia in craniofacial microsomia (CFM) with accompanying obstructive sleep apnea (OSA). Patients with severe mandibular hypoplasia require reconstruction functionality and esthetical features. In the cohort, reconstructions based on free fibular flaps (FFF) may be the most effective way. Patients aged 4–17 years with severe mandibular hypoplasia were treated with FFF, which initially improved the respiratory function assessed on polysomnography (AHI). In the next stages of treatment of cases with respiratory deterioration, it was indicated to perform distraction osteogenesis (DO) of the mandible and the structures reconstructed with FFF. All surgeries were planned in accordance with virtual surgery planning VSP. The aim of the study was to prospectively assess the effectiveness of multi-stage mandibular reconstruction in craniofacial microsomia with the use of a free fibula flap in terms of improving respiratory failure due to obstructive sleep apnea (OSA). The FFF reconstruction method, performed with virtual surgical planning (VSP), is proving to be an effective alternative to traditional methods of mandibular reconstruction in patients with severe CFM with OSA.

## 1. Introduction

The aim of the study was to retrospective assess the effectiveness of multi-stage mandibular reconstruction in craniofacial microsomia (CMF) with the use of a free fibula flap (FFF) in terms of improving respiratory failure due to OSA.

Craniofacial microsomia is the second most common congenital disorder of the head and neck. The occurrence of its severe form is rare. Mandibular hypoplasia is a feature of CFM. Other abnormalities include facial nerve palsy, ear anomalies, or facial soft tissue deficit on the same side [1]. The mandible is often the most functionally affected structure [2,3,4].

A possible treatment for mandibular hypoplasia in neonates and infants is mandibular distraction, but this is not possible when a mandibular bone is missing [5,6]. It becomes necessary to perform mandibular reconstruction, which enables further DO. Mandibular deficiencies can be corrected by bone grafting, distraction osteogenesis, or a combination of these methods [7]. In the case of patients with severe craniofacial microsomia, satisfactory correction is difficult to achieve with traditional methods. Most cases of CFM are well managed by conventional techniques, including costochondral grafts (CCG). Mandibular reconstruction is beneficial in CFM to address functional problems, such as airway and facial symmetry, mandibular and maxillary growth, and dental development. The application of conventional techniques may be limited in cases with severe mandibular displacement.

Free fibular flap reconstruction (FFFR) has been introduced as a new treatment option for patients with severe CFM. For patients with severe mandibular hypoplasia, FFFR may prove to be the most effective way of restoring mandibular shape and function [8]. The use of free tissue transfer for head and neck reconstruction has been proven to be safe and effective in the pediatric population [9,10,11].

The term obstructive sleep apnea (OSA) describes a syndrome of upper airway dysfunction during sleep that is characterized by increased upper airway resistance and the collapse of the throat. Symptoms of sleep apnea include snoring and/or increased work of breathing during sleep. Obstructive apnea includes many clinical entities of varying severities. It is characterized by snoring, labored breathing during sleep, and periods of complete or partial obstruction. Because OSA is associated with neurodevelopmental, metabolic, and cardiovascular consequences, an accurate diagnosis based on a patient examination and polysomnography (PSG) is important. Studies on the prevalence of OSA in patients with CFM have shown high variability, ranging from 7% to 67%. According to the authors, patients with severe face deformities are at risk for severe forms of OSA [12,13]. Children with various craniofacial conditions have been shown to be at increased risk for upper airway obstruction. The lack of prospective studies makes the prevalence of OSA and the causes of OSA in this population difficult to determine. OSA is the result of both structural factors that reduce airway size and neuromotor deficits that impair the patient's ability to maintain an open airway during sleep. Structural factors, such as a retracted and underdeveloped mandible, cause a reduction in the volume of the upper airway by shifting all structures backwards, closing the volume of the upper airway. Craniofacial diseases cause the dysfunction of the muscles of the mouth and throat, which affects swallowing, speech, and breathing. In children with craniofacial deformity, the ratio of length to tension of the muscles of the upper respiratory tract is changed, preventing them from working efficiently. One of the most common impairments in children with craniofacial disorders is feeding. This usually results in longer feeding times and can cause other problems, including malnutrition, dehydration, or aspiration of contents into the upper respiratory tract. Infants with craniofacial defects are at risk of poor growth, especially early in life. Sudden death during sleep may also occur in this group. Other symptoms of OSA in children include difficulty breathing while sleeping, drowsiness and night awakenings, learning disabilities, neurocognitive deficits, or failure to thrive. Obstructive sleep apnea is a serious disorder that manifests itself in excessive sleepiness. OSA is an independent risk factor for hypertension, ischemic heart disease, stroke, heart failure, atrial fibrillation, insulin resistance, and sudden death [14,15].

## 2. Materials and Methods

The records of patients who presented with severe CFM with mandible hypoplasia classified as severe, according to Pruzansky III, and who were treated were reviewed retrospectively (Table 1).

The inclusion criteria were related to the treatment protocol used at our department, for confirmed CFM with severe mandibular hypoplasia and the deterioration of OSA, as confirmed on examination and PSG (AHI). The treatment indications included deteriorating respiratory disorders confirmed by clinical symptoms and PSG. The factors qualifying for the primary reconstruction were severe mandible defect and worsening clinical symptoms and sleep parameters in PSG. The conservative definition of pediatric OSA [16] is AHI < 1 = normal, AHI 1–5.0 = mild, AHI 5.1–9.9 = moderate, and AHI > 10 = severe. The factors qualifying for mandibular distraction in the next stage included deteriorating clinical parameters and AHI on PSG [17].

The exclusion criteria included the absence of signed informed consent; a low body mass index; general health problems; no respiratory distress and central respiratory distress; disqualification for an anesthetic reason; disqualification for ENT reasons, such as tracheomalacia, laryngomalacia, or tonsil hyperplasia; and irregular or no reporting to requested follow-ups or the final follow-up.

The assessment of AHI was the basis for the qualification for surgery and evaluation of the surgery outcome. Patients who presented a decrease in AHI on follow-up PSG after mandibular FFFR required subsequent mandible DO.

Teenage patients were qualified for bimaxillary surgery, temporomandibular joint prosthesis, or both.

Patients qualified for mandibular FFFR underwent pre-operative planning. Surgical templates were prepared for bone cutting of both the mandibular defect and osteotomy of the fibula, as well as production stereolithographic models. Virtual planning and individual implants were used in accordance with manufacturer recommendations (CHM, Poland, Stare Juchy). The surgery involved FFF harvesting and FFF modifications, including the selection of the donor fibula, the site and the type of neck vessels used for anastomosis, the location of a skin island, and the preparation of segmental osteotomies of fibula, applied individually to each case.

A CT scan was performed before the surgery to prepare for virtual planning. Then a CT was carried out after the surgery to control and assess the correctness of the performed reconstruction. In the protocol, follow-up CT scans were performed 6 months after the surgery, before the removal of the stabilizing plates. No imaging examinations were performed in the following years. The next examination was performed just before the mandibular DO to assess the amount of bone and to plan the position of the distance device and the osteotomy line.

## 3. Results

The treatment of 13 patients with severe craniofacial microsomia was analyzed retrospectively.

Among the 13 patients, three started treatments at the age of 4, four at the age of 5, and then at the age of 7, 8, 9, 14, and 15 years, one patient at each age. The study group included five girls and eight boys. The patients’ follow-up period ranged from 15 to 77 months (mean 39.00). In seven patients, the mandible was deformed on the left side, and in the other six, on the right side. Before treatment, two patients underwent unsuccessful treatment attempts in other centers: these were free CCG. The first patient was 5 years old, and the other was 7 years old.

All patients underwent the reconstruction of the ramus and body of the mandible deficit with FFF microvascular flaps. In four patients, one fibular fragment was used, and in the other nine patients, two bone fragments of FFF were used. In four patients, no skin island was used, and in the remaining nine, a skin island was used to correct the deficit of soft tissues (Table 2; Figure 1 and Figure 2).

In the patients who additionally had a soft tissue flap, only two had a single-element bone fragment. In four patients, an additional unilateral sagittal split osteotomy was performed on the opposite side. These were patients in whom a two-component FFF was used to reconstruct the mandibular ramus and body. Two patients developed complications in the partial resorption of the bone graft, and in one patient, despite the healing of the skin island, complete resorption of the bone part of the graft occurred. Virtual planning with templates and individual implants was only not used in two patients. They were treated with single element FFF grafts. 

Six patients from the study group underwent DO after FFFR (Figure 3). In these patients, the bilateral DO of the mandibular body was performed. This was for increasing the advancement of the mandible. Patients eligible for DO were aged 6 (three patients), 8, 10, and 12 years. Four of these patients had tracheostomy before reconstructive treatment was started. The remaining patients had severe respiratory disorders confirmed by PSG, with high levels of AHI: above 20. In three patients, the tracheostomy was removed after FFFR. In one patient, the tracheostomy was not removed. This patient had a complication in the form of bone flap resorption; no distraction was performed. In the remaining patients, after FFFR, respiratory improvement was observed, and AHI decreased by 8 AHI units on average. 

Bilateral DO of the mandibular body was performed in six patients. DO was not performed in the other seven patients because the respiratory parameters were good and there were no indications for further treatment. One patient was a teenager (aged 15 years) and FFFR combined with orthognathic surgery was sufficient to correct the defect. DO was repeated in two patients later, correcting asymmetry and malocclusion. One patient underwent DO of the mandibular ramus to prepare for FFFR. Four patients had tracheostomy before FFFR treatment. The AHI before FFFR in non-tracheostomy patients was over 20 in all cases, with a mean of 20,0. After FFFR, the tracheostomy was removed in three patients. In the remaining patients, the AHI after FFFR ranged from 12 to 21 (mean 11.4). The patient with the highest AHI, 22, was a tracheostomy patient. 

During the growth of the patients, respiratory parameters deteriorated over time, reaching high AHI values. Indications for DO include exacerbating OSAS confirmed clinically and on PSG (AHI from 18 to 24, mean 21.5). The time between FFFR and DO was two years in four patients, three years in two patients, and five years in one patient. There was a two-year interval between FFFR and DO in younger children who underwent FFF reconstruction at the age of four. Subsequently, during follow-up examinations, respiratory parameters (AHI) deteriorated in 6 patients (AHI from 18 to 24, mean 21.5). These patients were qualified for mandibular DO to open the upper airways and improve breathing. The average AHI after distraction was 12 index. Improvements were noted in this context [18] (Table 3).

## 4. Discussion

The clinical evaluation of the patient for presumptive OSA is an essential first step in diagnosis. The assessment is based on the key symptoms of sleep apnea: snoring or witnessing a stop in breathing during sleep, insomnia, sleep hygiene, and the patient’s sleep schedule. Other symptoms, such as restless legs syndrome or parasomnias, should be excluded as possible contributing factors to the patient’s complaints. The physical examination for patients with suspected OSA was comprehensive and included the assessment of blood pressure, obesity index, and nose, throat, and craniofacial functions. The classification system for OSA in children is still being discussed and has not yet been standardized. Large observational studies of healthy children were conducted to define reference values for respiratory parameters during sleep [19]. Interestingly, reference values for common PSG parameters, such as AHI, do not follow a normal distribution. Tonsillectomy and adenoidectomy are the mainstay of pediatric OSA treatment. A small group of pediatric patients with OSA will still have respiratory disorders, which indicates a craniofacial etiology [20]. Continuous positive airway pressure (CPAP) is therefore indicated for treatment until surgical intervention can be performed to correct the skeletal abnormality that produces this disease. CPAP is relatively well tolerated in a large proportion of these patients, with studies reporting up to 80% adherence [21]. However, achieving good CPAP mask adherence in craniofacial pediatric patients is quite difficult. Additionally, it can be a potential limitation of CPAP in the pediatric population.

The most difficult and challenging patients are those with advanced mandibular hypoplasia causing respiratory distress and increasing asymmetry to decreasing saturation, leading to the need for tracheostomy. Many patients with mild to moderate mandibular deformities can be treated successfully with a combination of these techniques, but these procedures are often not sufficient to adequately reconstruct the severely hypoplastic mandible deformity. This makes it more challenging to treat younger patients with this type of deformity. The need for tracheostomy is still an important element affecting the patient’s development. On the other hand, if early reconstruction stimulated by osteogenic DO is not performed, the asymmetry aggravates and prevents successful reconstructions at a later age. FFFR has become an invaluable tool in mandibular reconstruction, and it has a significant advantage over bone grafting and DO in complex or severe mandibular hypoplasia [18]. Non-vascularized bone grafts, such as CCG and iliac crest bone grafts, have a high percentage of atrophy and failure [22]. These grafts show high rates of resorption or unpredictable growth patterns or may lead to ankylosis [23,24,25]. In addition to a poorly vascularized recipient bed, these patients have hypoplastic fibrotic soft tissues on the affected side, which induces high pressure on the inserted bone graft, also affecting faster resorption. The high failure rate of the free non-vascularized grafts in CFM is a consequence of defective surrounding soft tissues. In the case of the patients in our study group, all microsurgical FFF grafts were performed initially, and in contrast to the reported cases, two cases were two patients after multiple reconstructions with CCG grafts and one teenage patient not treated previously. CCGs are a choice for the reconstruction of the mandibulae. Cartilage is the center of growth in the literature; however, this growth is unpredictable and can range from no growth to hypertrophy. CCG can be taken at the age of about 10 years due to the development of the chest, which significantly delays the possibility of reconstruction and the planning of the next stages of treatment. It can also lead to damage of the upper respiratory tract by tracheostomy. In today's era of microsurgery, it is not too aggressive to harvest a fibula from a young patient. It is no less aggressive to take CCG and wait for the chest to grow, which delays the start of the patient’s respiratory rehabilitation.

DO of CCG has a high complication rate, up to 68% [26]. Complications include device failure, no healing, temporomandibular joint ankylosis, and the lack of consolidation, but DO of costochondral grafts is still possible [27]. The authors note that FFFs have a low resorption rate compared with non-vascularized bone grafts and are stable over time [28]. We no longer consider non-vascularized bone grafts to be the best option in this group of patients, and in recent cases, bone grafting was attempted before free flap surgery. It can be seen from the presented material that mandibular DO was intended to enlarge the upper airway, causing the reduction of OSAs which are destructive to children’s overall development. These patients, if untreated, would probably have been condemned to tracheostomy. In the case of two patients, permanent tracheostomy could be removed after reconstruction. This demonstrates the effectiveness of the method and the disappearance of the problem of respiratory disorders associated with severe mandibular hypoplasia.

The reconstruction of the mandible in CFM using different donor flaps has been reported. Mandibular reconstruction using scapular flaps to restore facial symmetry have been reported [29]. FFF is the predominantly used flap for mandibular reconstruction in children due to the ability to collect a large amount of bone, ease of preparation, ability to collect a simultaneous soft-tissue flap for soft-tissue reconstruction, low incidence of complications at the donor site, and ability to perform effective DO [30]. Additional benefits include the ability to perform multiple osteotomies without compromising blood supply and the use of septocutaneous perforators to obtain soft tissue for facial contour reconstruction. The anatomy of FFF allows the bone to be divided into multiple segments and a double bar created to achieve adequate alveolar height and anatomical contour. The volume of good quality bone in FFF allows for the placement of dental implants or TMR prostheses, which is crucial for obtaining proper dentition (Figure 4). Early FFF grafting allows for the improvement and reconstruction of the posterior facial height and successful mandibular DO, even in a multistage protocol, to catch up with the growth disorder at a later stage. It allows for the final reconstruction with allopathic joint implants. FFF can be used successfully in patients who have undergone multiple previous surgeries in the same surgical field, including failed bone grafts and DO or previous free flaps for soft tissue augmentation. There is no objective evidence for bone flap growth. Studies show continued mandibular growth of the residual natural mandible [31]. The analysis has shown that “growth potential” was observed in 58% of patients. Factors associated with improved growth potential include condylar preservation in reconstructions performed between 8 and 12 y.o., when there is a period of rapid mandibular growth [10].

The development of virtual surgical planning technology has resulted in significant advances in complex craniofacial surgery (Figure 5) [32]. Its use facilitates complex, multi-planar bone movements and allows the reconstructive surgeon to accurately predict postoperative anatomical relationships. Virtual preoperative planning also allows the precise coordination of resection and reconstruction. Prefabricated cutting instruments reduce surgery time and allow excellent bone contact between the mandibular bone fragments and the transferred bone, which contributes to better results. In the case of Pruzansky IIB, III mandibular reconstruction in patients with severe facial CFM, virtual surgical planning is particularly important. These patients have an asymmetric skull base and no mandibular fossa, making it very difficult to determine where the fibula should be located (Figure 5).

Our philosophy is that FFFR is mainly to open the upper airway. In the next step, it is used to prevent progressive asymmetry by aligning the mandibular position and rebuilding the height on the deficient side. In addition, the graft is a preparation for possible DO in the future.

In our material, mandibular DO was performed in six cases. In OSAS cases, patients warranted subsequent DO. Our cases involving DO around the age of 6 support this hypothesis. DO significantly improves the symmetry and conditions of the underlying soft tissues.

With time and follow-up, we will be better able to determine the optimal timing for performing mandibular microvascular reconstruction and subsequent DO, and finally possible alloplastic TMJ reconstruction. We can continue observing the development of patients and assessing the growth of the transplanted bone. We can monitor the assessment of the quality of life based on questionnaires and the final objective assessment of the treatment effects after reaching adulthood, based on the number of procedures performed, the number of complications, and the achieved effects of OSA reduction, and the effects of mandibular function, the possibility of orthodontic, and implant treatment, and finally aesthetic effects.

Perhaps the main advantage of employing FFFR is the ability for reproducible DO with associated benefits. As there is no mandibular fossa, the addition of a cartilaginous fibular head would still not result in the formation of the temporomandibular joint. We did not observe FFF growth, but we did note “remodeling” of the distal flap in the temporal bone zone and resorption in two cases. The remodeling of the distal part was observed in CT scans performed 6 months after surgery and in patients qualified for DO. Remodeling consisted of smoothing the sharp bone edges and shortening the distal part. One patient developed ankylosis between the fragment and the temporal bone, which was removed during the DO procedure.

The study was performed on a small group of patients due to the rarity of this type of pathology and indications for surgical intervention. Radiation protection of growing children is a limitation for CT assessment. On the other hand, clinical and polysomnographic examinations were performed every 6 months to find indications for possible DO. Observations will continue to be carried out to assess the effects of treatment and patients’ growth, especially after reaching maturity. Future areas of research include whether and by how much to overcorrect FFFR and when to perform DO. Finally, research is needed to answer the question of whether, after adequate bone aging in patients after FFFR and DO, TMJ reconstructive procedures will be needed.

## 5. Conclusions

The use of FFF may favorably affect the opening of the upper respiratory tract, reducing OSA. Some cases require subsequent DO. The continuous observation and assessment of apnea in this group of patients is necessary. It can be concluded that a hypoplastic mandible in CFM can be treated with good results. The FFF method, performed with virtual surgical planning, is proving to be an effective alternative to more traditional methods of mandibular reconstruction. In the cases of mandibular hypoplasia, the mandibular microvascular reconstruction with FFF can be considered a primary reconstruction modality. The FFF flap may serve as excellent material for subsequent DO, as well as bimaxillary surgery as the final stage of the treatment. FFF reconstruction with subsequent DO significantly improves respiratory function in patients with CFM.

## Figures and Tables

**Figure 1 jcm-12-01124-f001:**
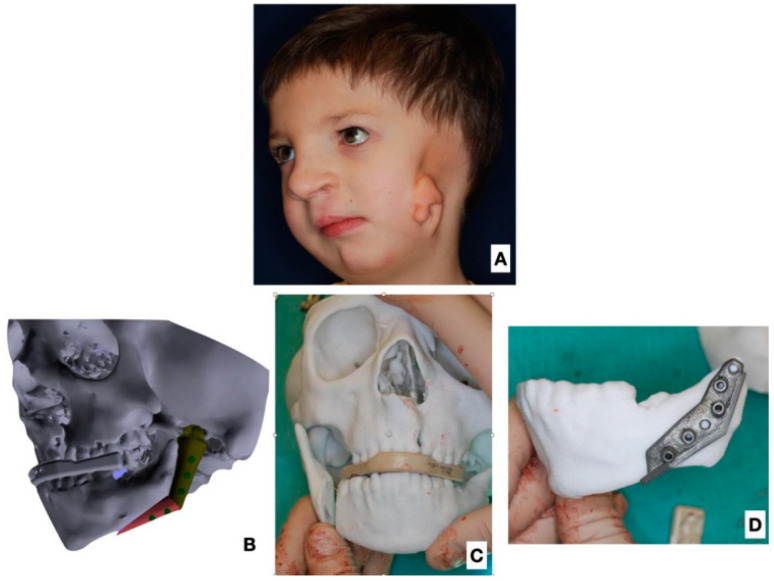
Patient 5 years old with HF Pruzansky III. (**A**) Three-quarter view, soft tissue deformity visible, with auricle microtia. (**B**) 3D image reconstruction with planned free fibular flap reconstruction. (**C**) Stereolithographic model for intraoperative planning and control. (**D**) Stereolithographic model with intraoperative template.

**Figure 2 jcm-12-01124-f002:**
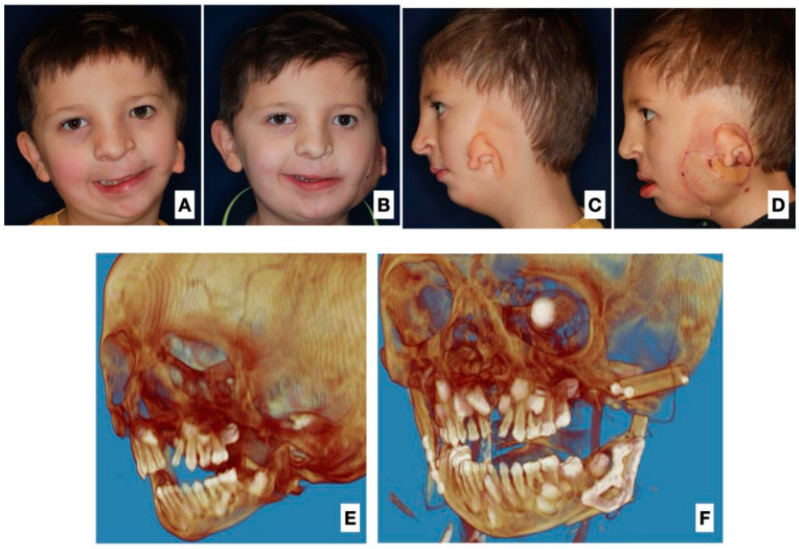
(**A**) Frontal view before surgery, showing asymmetry and soft tissue deficit. (**B**) View one month after surgery with visible correction of asymmetry and soft tissue deficit. (**C**) Side view before surgery. (**D**) Side view one month after surgery, visible insertion in cheek area. (**E**) Preoperative 3D image reconstruction. (**F**) 3D image reconstruction after surgery and mandibular FFF reconstruction.

**Figure 3 jcm-12-01124-f003:**
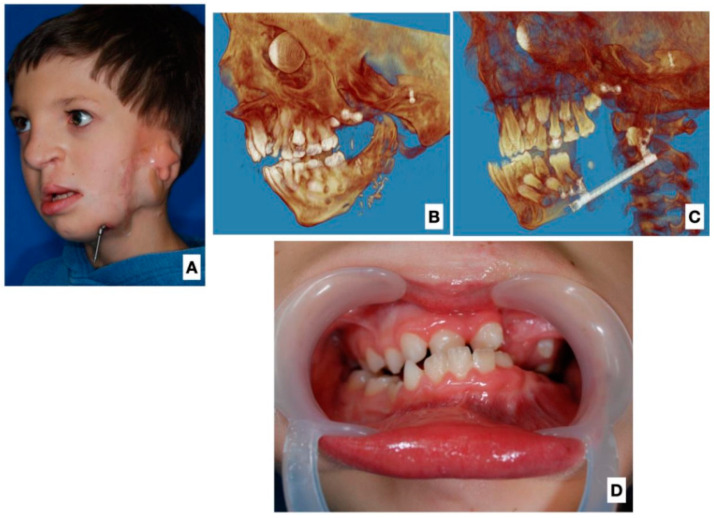
The same patient two years after surgery. (**A**) Three-quarter view after completion of osteodistraction. (**B**) 3D image reconstruction after FFF reconstruction. (**C**) 3D image reconstruction after completion of mandibular distraction. (**D**) Intraoral view with visible overcorrection and dentition set in reverse occlusion.

**Figure 4 jcm-12-01124-f004:**
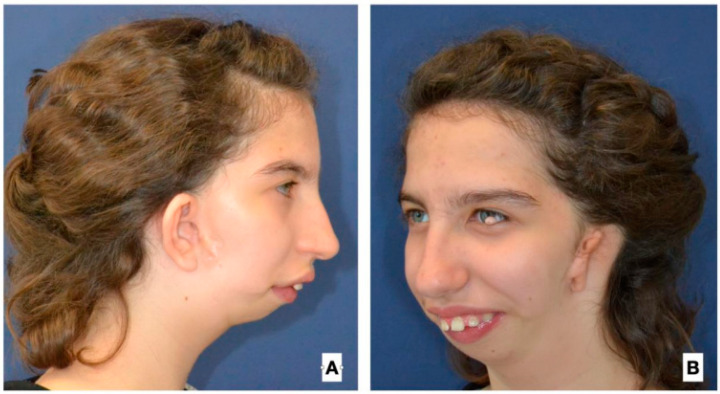
Patient aged 15 years with bilateral HFM. (**A**) Side view. (**B**) Three-quarter view.

**Figure 5 jcm-12-01124-f005:**
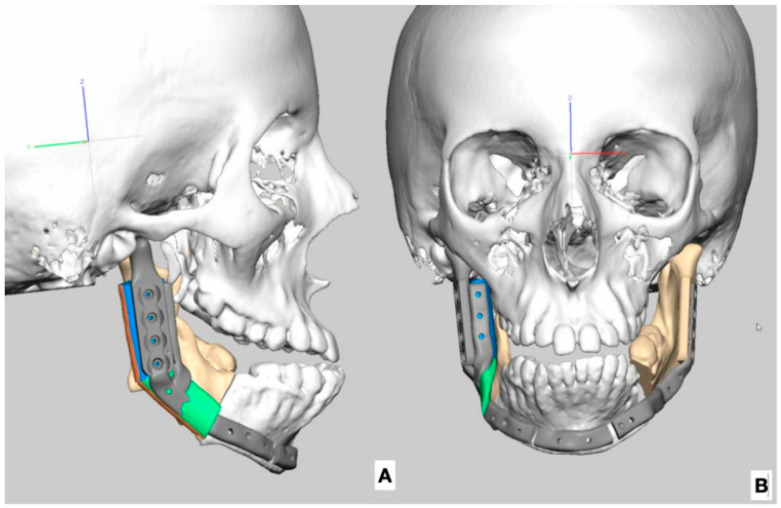
3D image reconstruction. (**A**) Plan of FFF mandibular ramus reconstruction in combination with TMJ alloplastic prosthesis. (**B**) Frontal view of the plan of FFF mandibular ramus reconstruction, planned additional mandibular osteotomy on the left side.

**Table 1 jcm-12-01124-t001:** Characteristics of patients with HFM based on age, gender, side of deformity, determination of mandibular deformity based on Pruzansky classification, and determination of additional facial and general developmental deformities.

Age at 1st Operation y.o.	Gender	Side	Pruzansky	Additinal Deformations of Face	Other Deformations
4	Female	Left	III	Soft tissue atrophy	
4	Male	Left	III	Microtia, atresia, VII palsy, soft tissue atrophy	
4	Female	Left	III	Microtia, atresia, VII palsy, soft tissue atrophy, orbital dystopia, cervical spinal deformation	
5	Male	Left	III	Cleft LP orbital dystopia anoftalmia, mictai, VII palsy, soft tissue atrophy	Cervical spinal deformation, scoliosis
5	Female	Right	III	Microtia, atresia, VII palsy, soft tissue atrophy	
5	Male	Left	IIB	Microtia, atresia, VII palsy, soft tissue atrophy	Forearm deformation, cervical spinal deformation
5	Female	Right	IIB	Microtia, atresia, VII palsy, soft tissue atrophy	Forearm deformation
7	Male	Right	III	Microtia, atresia, VII palsy, soft tissue atrophy, orbital dystopia, deformation Cleft Tessier 11/4	Cervical spinal deformation
8	Male	Left	III	Microtia, atresia, VII palsy, soft tissue atrophy, orbital dystopia, cervical spinal deformation	
9	Male	Left	IIB	Microtia, atresia, VII palsy, soft tissue atrophy	
14	Male	Right	IIB	Microtia, atresia, VII palsy, soft tissue atrophy	
15	Female	Right	Bilat III/II B	Microtia, atresia, soft tissue atrophy	
17	Male	Right	III	Microtia, atresia, VII palsy, soft tissue atrophy	

**Table 2 jcm-12-01124-t002:** Characteristics of free flaps used for mandibular reconstruction in patients with HFM with definition of the affected side, number of bony elements of the fibular flap, collection of the skin island, reconstruction of the mandibular anatomical region, use of additional osteotomies during mandibular reconstruction with a free flap based on microvascular anastomoses, occurrence of complications during and after surgery, use of virtual planning and individual implants (VSP, IPS), follow-up period after reconstruction. CCG—costochondral graft, FFF—free fibula flap.

Age before Treatment y.o.	Previous Procedures	Side	Type of Flap	Number of Pieces	Soft Tissue Island	Complications	VSP and IPS	Time of Observation in Months
4		left	FFF	2	Yes	No	No	59
4		right	FFF	1	No	Bone resorption	No	77
4		right	FFF	1	Yes	No	Yes	15
5		left	FFF	2	Yes	No	Yes	9
5		left	FFF	1	Yes	No	Yes	17
5	CCG	left	FFF	2	No	No	Yes	24
5		right	FFF	2	Yes	Partial bone resorption	Yes	24
7		right	FFF	1	No	No	Yes	33
8		left	FFF	2	Yes	No	Yes	37
9		left	FFF	2	Yes	No	Yes	43
14		left	FFF	2	Yes	No	Yes	47
15		right	FFF	2	Yes	No	Yes	53
17	CCG	right	FFF	2	No	No	Yes	69

**Table 3 jcm-12-01124-t003:** Analysis of OSA based on AHI in patients undergoing FFFR, after fibular flap reconstruction. FFFR—free fibula flap reconstruction, TMR—temporo mandibular reconstruction alloplastic prosthesis, BIMAX—orthognathic bimaxillary surgery, TRACHEO—tracheostomy, DO—distraction osteogenesis, n—no tracheostomy, y—yes tracheostomy.

Age before Treatment y.o.	TRACHEO before FFFR	Tracheostomy Removal	AHI before FFFR	AHI after FFFR	AHI before DO	Age during DO y.o.	AHI after DO	Additional Surgery
4	n	n	26	16	22	6	10	
4	n	n	24	15	23	6	13	
4	y	y	TRACHEO	16	22	6	12	
5	n	n	23	11	18	8	16	
5	y	n	TRACHEO	21	24	8	12	
5	n	n	20	19	18	No DO	No DO/9	
5	y	y	TRACHEO	13	17	No DO	No DO/10	
7	y	n	TRACHEO	TRACHO	TRACHO	No DO	TRACHEO	
8	n	n	20	12	20	13	12	
9	n	n	17	10	15	No DO	No DO/9	
14	n	n	16	12	18	No DO	No DO/11	TMR/BIMAX
15	y	y	27	11	16	No DO	No DO/11	TMR/BIMAX
17	n	n	20	12	16	No DO	No DO/11	TMR/BIMAX

## Data Availability

The data are available upon request from the corresponding author.

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
