# Peer review of "Use of a Fibula Free Flap for Mandibular Reconstruction in Severe Craniofacial Microsomia in Children with Obstructive Sleep Apnea"

_jcm, 2023, doi:10.3390/jcm12031124_

Round 1

Reviewer 1 Report

This paper is focused on an interesting topic and it may be interesting for the readers. I want to compliments with the Authors for the work done and for the excellent iconography. Nonetheless, major revisions are needed in order to substantially improve it and a thorough English revision is required as well.   INTRODUCTION P1 L33: "prospectively": incorrect, as the paper has a retrospective design.    At the end of the introduction, it would be helpful for the reader to explain why "severe face deformities are at risk of sever forms of OSA" and how they manifest and therefore what are the meaning and the importance of this study. In my opinion, it is crucial for the reader to understand this key-passage.     MATERIALS AND METHODS   Staging of the patients according to OMENS classification is not described and it would be necessary.   P4L88: "assessment of AHI was the basis". It is "the basis", but the preoperative evaluation protocol was not described here. It must be precised. P4L89: "presented decrease in AHI". What is the amount of the decrease? What if it was not reached?   P4L91: "Teenage patients were qualified... where necessary". When was it necessary? How was assessed? Which criteria and tools were used?   P4L97: "The surgery involved FFF harvesting in accordance with standards". Which standards? Many harvesting techniques are described. Be more specific, please.   P4L107: In my opinion, this is the most important missing part. The follow-up protocol is not described at all. When were the re-evaluation of the OSAS programmed? When was the last follow-up examination? How often were the clinical and/or instrumental examinations performed?     RESULTS P8L139: Were the soft tissue defects intra- or extra-oral?   P9L148: "BSSO". How can it be bilateral if the sagittal split osteotomy was performed on the opposite side?   P9L150: "two patients developed complications". Only one patient is indicated in Table 2. Moreover, what was the flap failure rate?   At the end of the Results section, a better description of the follow-up results is required as explained above in the comment on the Materials and Methods section.     DISCUSSION P12L293: "we will be better able to determine the optimal timing". How do you plan to do that?   P13L299: "we did note remodeling of the distal flap". It was said that only one CT after six months was performed. What about subsequent follow up? This aspect is unclear and should be revised.   Limitations of the study are not discussed at all.   When added, discussion of the follow-up protocol would be necessary as well.     Finally, a careful English revision is required. Moreover, there is plenty of use of acronyms that are not previously specified in the text (OSA, DO, CCG, CPAP): this aspect should be revised as well.

Author Response

INTRODUCTION 

P1 L33:  "prospectively": incorrect, as the paper has a retrospective design. At the end of the introduction, it would be helpful for the reader to explain. CORRECTED

why severe face deformities are at risk of sever forms of OSA and how they manifest and therefore what are the meaning and the importance of this study. CORRECTED

MATERIALS AND METHODS   

Staging of the patients according to OMENS classification is not described and it would be necessary.

Nie może być OMENS, ponieważ nie analizujemy materiału na podstawie tej klasyfikacji. Zacytowanie OMENST w pracy zmusiłoby do zmiany wszystkich badań i całego materiału, a artykuł nie ma na celu określenia deformacji w OMENS klasyfikacji.

P4L88: "assessment of AHI was the basis". It is "the basis", but the preoperative evaluation protocol was not described here. It must be precise.

The inclusion criteria were related to the treatment protocol, used at our department, for confirmed CFM with severe mandibular hypoplasia and deterioration of OSA as confirmed on examination and PSG (AHI). The treatment indications included deteriorating respiratory disorders confirmed by clinical symptoms and PSG. The factors qualifying for the primary reconstruction were severe mandible defect and worsening clinical symptoms and sleep parameters in PSG. The conservative definition of Pediatric OSA [16] is AHI < 1 = normal, AHI 1–5.0 = mild, AHI 5.1–9.9 = moderate, AHI > 10 = severe. The factors qualifying for mandibular distraction in the next stage included deteriorating clinical parameters and AHI on PSG.[17]

P4L89: "presented decrease in AHI". What is the amount of the decrease? What if it was not reached?

Pacjent nie był kwalifikowany do DO

P4L91: "Teenage patients were qualified... where necessary". When was it necessary? How was assessed? Which criteria and tools were used?   

CORRECTED

P4L97: "The surgery involved FFF harvesting in accordance with standards". Which standards? Many harvesting techniques are described. Be more specific, please.   

CORRECTED

P4L107: In my opinion, this is the most important missing part. The follow-up protocol is not described at all. When were the re-evaluation of the OSAS programmed? When was the last follow-up examination? How often were the clinical and/or instrumental examinations performed?

Patients who presented decrease in AHI on follow-up PSG after mandibular FFFR required subsequent mandible DO.

A CT scan was performed before the surgery to prepare for virtual planning. Then an CT was done after the surgery to control and assess the correctness of the performed reconstruction. In the protocol, follow-up CT scans were performed 6 months after the surgery, before the removal of the stabilizing plates. No imaging examinations were performed in the following years. The next examination was performed just before the mandibular DO to assess the amount of bone and to plan the position of the distance device and the osteotomy line.

Subsequently, during follow-up examinations, respiratory parameters (AHI) deteriorated in 6 patients (AHI from 18 to 24, mean 21.5). These patients were qualified for mandibular DO to open the upper airways and improve breathing. The average AHI after distraction was 12,0. Improvements were noted in this context.

RESULTS

P8L139: Were the soft tissue defects intra- or extra-oral?

Additional benefits include the ability to perform multiple osteotomies without compromising blood supply and the use of septocutaneous perforators to obtain soft tissue for facial contour reconstruction.

P9L148: "BSSO". How can it be bilateral if the sagittal split osteotomy was performed on the opposite side? CORRECTED

P9L150: "two patients developed complications". Only one patient is indicated in Table 2. Moreover, what was the flap failure rate?   At the end of the Results section, a better description of the follow-up results is required as explained above in the comment on the Materials and Methods section.

Podane są dwie komplikacje resorpcja kości i częściowa resorpcja kości

DISCUSSION 

P12L293: "we will be better able to determine the optimal timing". How do you plan to do that? CORRECTED

P13L299: "we did note remodeling of the distal flap". It was said that only one CT after six months was performed.

What about subsequent follow up? This aspect is unclear and should be revised.  

Limitations of the study are not discussed at all.  When added, discussion of the follow-up protocol would be necessary as well. CORRECTED

Finally, a careful English revision is required. Attached is the Translation Agency's certificate of verification by a native speaker.

Moreover, there is plenty of use of acronyms that are not previously specified in the text (OSA, DO, CCG CORRECTED, CPAP CORRECTED): this aspect should be revised as well.

This is a retrospective study describing a multi-stage protocol for the management of severe mandibular hypoplasia in craniofacial microsomia (CFM) with accompanying obstructive sleep apnea (OSA).

In cohort, reconstructions based on free fibular flaps (FFF) may be the most effective way.

The aim of the study was to prospectively assess the effectiveness of multi-stage mandibular reconstruction in craniofacial microsomia with the use of a free fibula flap in terms of improving respiratory failure due to obstructive sleep apnea (OSA).

In the next stages of treatment of cases with respiratory deterioration, it was indicated to perform distraction osteogenesis (DO) of the mandible and the structures reconstructed with FFF.

Reviewer 2 Report

Thank you for giving me the chance to evaluate the article titled “Use of a fibula free flap for mandibular reconstruction in severe craniofacial microsomia in children with obstructive sleep apnea”. There are some points that I would like to share with the authors.

Keywords should be included in the article.

Although free fibula flap is a good option for treatment in these patients, I think that it is a radical surgical procedure and the surgeon should have special surgical experience. This situation should be mentioned as the disadvantage of the technique in the article. Distraction osteogenesis has been discussed by the authors as an alternative. “Management of obstructive sleep apnea in a Treacher Collins syndrome patient using distraction osteogenesis of the mandible. Doi: I think the case report 10.5125/jkaoms.2016.42.6.388” fits the purpose of the study and can be a good example. This report should be attached to the references.

A figure can be added regarding the Pruzansky classification.

I think reference 23 needs to be reviewed. The situation that does not comply with the spelling rules should be corrected.

The success of the treatment has been proven by the postoperative polysomnographic measurements of the patients. I think that well presented cases from plan to surgery will be a guide for other surgeons.

Best regards.

Author Response

Keywords should be included in the article. CORRECTED

Although free fibula flap is a good option for treatment in these patients, I think that it is a radical surgical procedure, and the surgeon should have special surgical experience. This situation should be mentioned as the disadvantage of the technique in the article. Distraction osteogenesis has been discussed by the authors as an alternative. “Management of obstructive sleep apnea in a Treacher Collins syndrome patient using distraction osteogenesis of the mandible. Doi: I think the case report 10.5125/jkaoms.2016.42.6.388. CORRECTED

I think reference 23 needs to be reviewed. The situation that does not comply with the spelling rules should be corrected. CORRECTED

Round 2

Reviewer 1 Report

The authors have properly corrected the paper and answered to the concerns raised, therefore I recommend to accept this paper for publication.

I want to compliment again with the Authors for the brilliant work done.